# Subjective Happiness, Frequency of Laughter, and Hypertension: A Cross-Sectional Study Based on the Japan Gerontological Evaluation Study (JAGES)

**DOI:** 10.3390/ijerph20095713

**Published:** 2023-05-03

**Authors:** Fumikazu Hayashi, Yuka Shirai, Tetsuya Ohira, Kokoro Shirai, Naoki Kondo, Katsunori Kondo

**Affiliations:** 1Department of Epidemiology, Fukushima Medical University School of Medicine, 1 Hikariga-oka, Fukushima 960-1295, Japan; 2Radiation Medical Science Center for the Fukushima Health Management Survey, Fukushima Medical University, 1 Hikariga-oka, Fukushima 960-1295, Japan; 3Department of Public Health, Graduate School of Medicine, Osaka University, Osaka 565-0871, Japan; 4Department of Social Epidemiology, Graduate School of Medicine, Kyoto University, Kyoto 606-8501, Japan; 5Department of Social Preventive Medical Sciences, Center for Preventive Medical Sciences, Chiba University, Chiba 263-8522, Japan; 6Department of Gerontological Evaluation, Center for Gerontology and Social Science, National Center for Geriatrics and Gerontology, Aichi 474-8511, Japan

**Keywords:** subjective happiness, laughter, hypertension, Japan, older adults

## Abstract

In recent years, positive psychological factors, such as subjective happiness and laughter, have been reported to be associated with cardiovascular disease. In this study, we examined the relationship of hypertension with subjective happiness and frequency of laughter using the data from the Japan Gerontological Evaluation Study (JAGES). Of the 138,294 respondents, 26,368 responded to a version of the self-administered questionnaire that included a question about the frequency of laughter in the JAGES 2013. In total, 22,503 (10,571 men and 11,932 women) were included in the analysis after excluding those with missing responses regarding a history of hypertension, frequency of laughter, and subjective happiness. The prevalence of hypertension in this study was 10,364 (46.1%). Multivariate logistic regression analysis showed that age, female sex, obesity, infrequent chewing, former and current drinker, seeing three to five friends, and the absence of hobbies were positively associated with hypertension. However, infrequent laughter/high level of subjective happiness, frequent laughter/high level of subjective happiness, being underweight, and current smoker were negatively associated with hypertension. As per the findings of this study, it was determined that subjective happiness was negatively associated with hypertension. Therefore, this study suggests that having more opportunities to feel happiness may be important in preventing hypertension.

## 1. Introduction

Subjective happiness has been recognized as a factor that can positively impact the cardiovascular system. In recent years, studies in Western countries have shown a reduced incidence of cardiovascular disease and hypertension in individuals with higher levels of subjective happiness and emotional vitality [1,2,3]. Additionally, some studies have indicated that emotional vitality, positive emotions, and life purpose, which are considered components of happiness, can lower the risk of stroke [4,5,6]. Hence, enhancing subjective happiness could benefit the cardiovascular system and, in turn, improve overall health. However, a report indicated that the correlation between happiness and chronic diseases, including cerebral infarctions and heart attacks, is inconsistent and further investigation is necessary [7].

Recently, laughter has received attention as a factor with a positive effect on the cardiovascular system. People with frequent laughter have been reported to have a lower risk of cardiovascular disease [8,9]. A study using data from the Japan Gerontological Evaluation Study (JAGES) 2013 found an association between the frequency of laughter and the prevalence of cardiovascular disease [10]. Other studies suggest that laughter interventions may bring many health benefits, such as increased heart rate variability [11] and improved immune system [12,13], HbA1c [14], and vascular function [15,16]. These findings suggest that having more opportunities for laughter in daily life may benefit the cardiovascular system and contribute to improved health.

Previous studies examining the correlation between laughter and its health benefits have predominantly focused on the following two types of laughter: spontaneous and intentional. Spontaneous laughter has positive effects, whereas intentional laughter—also known as simulated laughter—is devoid of emotional components. Argyle hypothesized that positive emotions stemming from laughter may affect one’s health status [17]. Conversely, a laughter yoga intervention study utilizing simulated laughter reported that laughter had health benefits. Notably, it reduced depression scores and body mass index [18,19]. Therefore, whether positive emotions from laughter have health benefits remains a point of contention.

A review of the psychological implications of laughter has shown that laughter promotes psychological wellness [20]. However, the impact of laughter frequency and the difference in cardiovascular disease prevalence between patients with high and low subjective well-being remains unclear. Currently, no study has assessed the connection between laughter and subjective happiness with cardiovascular disease. While subjective happiness is a psychological factor, laughter can occur without emotions and can be considered a separate behavioral factor. Thus, this study aimed to examine the relationship between hypertension, which is considered to have a high prevalence among cardiovascular diseases, and the frequency of laughter and subjective happiness, using data from the JAGES. In the current study, individuals who frequently laugh and who have high levels of subjective happiness were hypothesized to show a low prevalence of hypertension.

## 2. Materials and Methods

### 2.1. Study Group

A flow chart of this study is shown in Figure 1. The JAGES started in 2010 to create a scientific basis for preventive policies for a healthy and long life. The JAGES cohort was established in 2010 to investigate factors affecting the health of independent individuals over 65 in 30 municipalities in Japan. In 2013, the JAGES mailed questionnaires to 195,290 residents over 65: 138,294 responded (70.8%). The study used data from 26,368 respondents who answered version B, which assessed oral hygiene, optimism, subjective health, and frequency of laughter. Finally, 22,503 respondents (10,571 men and 11,932 women) were included after excluding those missing information on hypertension, frequency of laughter, and subjective happiness. The study analyzed the association of hypertension with frequency of laughter and/or subjective happiness along with factors such as age, sex, height, weight, caregiving, household income, number of teeth, chewing ability, drinking habits, smoking habits, the Tokyo Metropolitan Institute of Gerontology Index of Competence (TMIG-IC), frequency of visiting friends, family relationships, work status, hobbies, and marital status.

### 2.2. Definitions of Hypertension

Respondents who reported having hypertension as a current or past medical condition were defined as having hypertension.

### 2.3. Definitions of Laughter/Subjective Happiness

Laughter was measured on a 4-point scale, with “almost every day” being defined as “frequent laughter” and the other responses as “infrequent laughter”. Subjective happiness was rated on a scale of 1 to 10, with scores ≥8 being defined as “high level of subjective happiness” and scores <8 as “low level of subjective happiness”. The combination of laughter and subjective happiness was divided into four categories: “infrequent laughter/low level of subjective happiness”, “frequent laughter/low level of subjective happiness”, “infrequent laughter/high level of subjective happiness”, and “frequent laughter/high level of subjective happiness”.

### 2.4. Definitions for Obesity and Underweight

Body mass index (BMI) has been shown to be linked with clinic systolic and diastolic blood pressures [21]. According to the World Health Organization, obesity is defined as a BMI of ≥30 kg/m^2^ [22]. However, only 2–3% of Japanese adults have a BMI ≥30 kg/m^2^, which is much lower than in Western countries [23]. In this study, BMI was calculated from self-reported height and weight, and obesity was defined as a BMI ≥ 25 kg/m^2^ (per the recommendation of the Japan Society for the Study of Obesity) [24]. Underweight was defined as a BMI < 18.5 kg/m^2^.

### 2.5. Definition of Caregiving and Assistance Status

Tamada et al. reported that people who rarely laugh have a higher risk of death or need for care than those who laugh daily [25]. Caregiving/assistance status was scored using a 3-point scale: “not needed”, “needed but not received”, and “received”.

### 2.6. Definition of Household Income

The Multi-centre National Population Health Examination Surveys (WOBASZ) study in Poland found a correlation between household income and hypertension prevalence [26]. The household income was rated on a 15-point scale, with a range from JPY < 0.5 million to JPY ≥ 12 million. Low income was defined as JPY < 2 million based on the 2014 National Health and Nutrition Survey in Japan [27].

### 2.7. Definitions Related to the Number of Teeth and Chewing Ability

Several studies have linked hypertension with periodontal disease [28]. The oral environment is thus an important factor in hypertension studies. In this study, the number of teeth was assessed using a 5-point scale: 0, 1–4, 5–9, 10–19, and ≥20. The report of the 2013 National Health and Nutrition Examination Survey in Japan showed that a higher percentage of people with ≥20 teeth can eat anything compared to those with <20 [29]. Thus, “≥20 teeth” and “<20 teeth” were compared. The hardness of food that could be eaten was also scored on a 5-point scale: “anything”, “almost”, “limited”, “rarely chewing”, and “liquid diet”.

### 2.8. Definition of Drinking and Smoking Habits

Heavy drinking and smoking positively impact blood pressure [30,31]. Drinking habits were scored on a 3-point scale: “no drinker”, “former drinker”, and “current drinker”. Smoking habits were also scored on a 3-point scale: “no smoker”, “former smoker”, and “current smoker”.

### 2.9. Definition of TMIG-IC

Activities of daily living have been reported to be associated with hypertension [32]. The TMIG-IC was used to evaluate BADLs, such as walking, moving, eating, bathing, toileting, and dressing [33]. A score of 11 in the 65–74 age group or ≥10 in the 75+ age group was considered “good” BADL and a score below that was considered “poor” BADL.

### 2.10. Definitions Regarding Social Network

Kim et al. found a link between small social networks and metabolic syndrome [34]. A 6-pt scale measured frequency of seeing friends: “4+ times/week”, “2–3 times/week”, “once/week”, “1–3 times/month”, “a few times/year”, and “not seeing friends”. A 5-pt scale measured number of friends met: “≥10”, “6–9”, “3–5”, “1–2”, and “0 (none)”. A 3-point scale determined family living arrangement: “living alone”, “with family”, and “others”. Results were divided into “living alone” and “others”. Employment was determined using a 2-point scale: “employed” and “unemployed”. Hobbies reflect participation in self-centered social networks [35]. A 2-point scale determined hobby presence: “yes” or “no”. A 5-point scale was used to determine marital status: “with spouse”, “bereaved”, “divorced”, “never married”, and “others”. Respondents were divided into “spouse” and “others” groups.

### 2.11. Statistical Analysis Methods

The statistical software SAS 9.4 (SAS Institute Inc., Cary, NC, USA) was used to analyze data and present results as mean ± standard deviation or number (%). Unpaired *t*-test and χ^2^ test were used to examine the association between hypertension and each item. Logistic regression models were used to obtain age- and sex-adjusted ORs and 95% CIs to identify factors associated with hypertension. Factors adjusted included age, sex, laughter, happiness, BMI, chewing ability, drinking habits, smoking habits, friend visits, hobbies, and marital status. ORs were also obtained for men and women separately, and multivariate logistic regression models were used for laughter and subjective happiness subanalyses. All tests were two-tailed, and *p* < 0.05 was considered statistically significant.

## 3. Results

The study included 22,503 subjects with a prevalence of hypertension of 46.1% (10,364 subjects). The results showed that subjects with hypertension were older than those without hypertension (Table 1). Hypertension was associated with factors such as laughter/subjective happiness, BMI, assistance status, number of teeth, chewing ability, drinking habits, smoking habits, number of friends met, marital status, employment status, and hobbies (Table 1).

Therefore, age- and sex-adjusted logistic regression analysis was conducted using factors that showed significant differences in the unpaired *t*-test and χ^2^ test (Table 2). As per the age- and sex-adjusted logistic regression analysis, age, female sex, obesity, infrequent chewing, former drinker, current drinker, no smoker, three to five friends met, and no hobbies were positively associated with hypertension, whereas infrequent laughter/high level of subjective happiness, frequent laughter/high level of subjective happiness, being underweight, and current smoker were negatively associated with hypertension.

Next, multivariate logistic regression was conducted using factors that showed significant differences in the age- and sex-adjusted logistic regression model (Table 2). As per the results, age, female sex, obesity, infrequent chewing, former drinker, current drinker, three to five friends met, and no hobbies were positively associated with hypertension. Meanwhile, infrequent laughter/high level of subjective happiness, frequent laughter/high level of subjective happiness, being underweight, and current smoker were negatively associated with hypertension.

Multivariate logistic regression was conducted separately for men and women to examine differences. In men, age (1.02, 95% CI: 1.01–1.03), obesity (1.93, 95% CI: 1.76–2.13), former drinker (1.41, 95% CI: 1.21–1.65), and current drinker (1.78, 95% CI: 1.63–1.94) had a positive association with hypertension. Infrequent laughter/high level of subjective happiness (0.85, 95% CI: 0.77–0.94), being underweight (0.44, 95% CI: 0.36–0.54), and current smoker (0.77, 95% CI: 0.69–0.86) had a negative association. Frequent laughter/high level of subjective happiness had a negative trend toward hypertension (0.90, 95% CI: 0.81–1.00). In women, age (1.05, 95% CI: 1.04–1.06) and obesity (2.19, 95% CI: 1.99–2.41) had a positive association with hypertension, and being underweight (0.54, 95% CI: 0.46–0.62) had a negative association.

Finally, multivariate logistic regression analysis was performed separately for laughter frequency and subjective happiness (Table 3). No significant association was found between frequent laughter and hypertension (OR: 0.98, 95% CI: 0.93–1.04). However, higher subjective happiness was found to significantly lower the risk of hypertension compared with lower happiness (OR: 0.91, 95% CI: 0.86–0.97).

## 4. Discussion

Several studies have shown a link between positive emotions and a reduced risk of hypertension [3,36,37]. This study also found that individuals with higher levels of subjective happiness had a lower incidence of hypertension compared with those with lower levels of happiness. One explanation for this relationship is that higher subjective happiness is linked to lower stress levels. Cortisol, a hormone associated with stress, increases with stress through the hypothalamic–pituitary–adrenal system. Men with higher subjective happiness have lower cortisol levels in their saliva, regardless of their employment status [38]. Additionally, interventions aimed at inducing a positive mindset decrease cortisol levels and reduce the cortisol response to stress [39]. These findings suggest that subjective happiness may help prevent hypertension by reducing cortisol levels and the associated stress response.

In the present study, regardless of laughter frequency, subjective happiness was negatively associated with hypertension in men. This finding could be explained by the difference in the level of happiness between men and women. The percentage of women (54.6%) with a high level of subjective happiness was significantly higher than that of men (47.1%) (*p* < 0.001) (Data not shown). Using the Subjective Happiness Scale (SHS) [40] to assess subjective happiness among Japanese people in their 20s–70s, older age groups were found to have a higher subjective happiness than younger ones. Furthermore, subjective happiness was higher in people with a spouse than in those without. Finally, women had a higher subjective happiness than men [41]. Moreover, happiness was inversely related to cortisol levels and heart rate as assessed using the gait monitoring method [42]. Furthermore, a high heart rate can be a predictor of mortality and cardiovascular disease risk according to prospective epidemiologic studies. Moreover, a consistent association was observed between mortality and heart rate in men [43]. These findings suggest that men are more susceptible to increased secretion of stress hormones and decreased cardiovascular function due to a lower subjective happiness than women. Promoting subjective happiness, at least for men, may be important to prevent cardiovascular disease.

In this study, infrequent social interaction and the absence of hobbies were positively associated with hypertension. Ngamaba KH has reported a positive association between subjective well-being and trust in other people, importance of friends and family, leisure activities, and weekly attendance to religious services [44]. In adults and older adults living in the rural areas of Japan, less television viewing and more mentally active sedentary behaviors (e.g., talking to others, engaging in hobbies) were associated with greater happiness [45]. Ikigai may promote health and well-being among older Japanese adults [46]. These reports suggest that hobbies may improve subjective well-being. A cross-sectional study using JAGES 2016 data showed that community-level civic engagement was negatively related to hypertension [47]. The current study showed that individuals with hobbies were more likely to be socially active and have more opportunities to interact with others compared with those without hobbies. Therefore, the number of social interactions and hobbies may be an indirect measure of social interaction. These results suggested that some hobbies might help prevent hypertension in older adults by promoting well-being and communicating with others.

Frequent laughter and high levels of subjective happiness were found to have a negative association with hypertension compared with infrequent laughter and lower subjective happiness. However, the results showed that even with frequent laughter, if subjective happiness was low, it was not related to hypertension. A separate analysis of laughter frequency alone also showed no significant link to hypertension. Despite this, previous studies have reported a higher risk of cardiovascular disease among individuals with infrequent laughter [10] and increased blood pressure in men who laughed less than once a week [48]. Additionally, Low et al. conducted a study among nursing home residents and found that those who received weekly humor therapy for 9–12 weeks had fewer symptoms of agitation and increased happiness compared to the control group [49]. Another study by Ghodsbin et al. reported that 90 min of laughter therapy twice a week for 6 weeks improved anxiety, insomnia, and physical symptoms in community residents aged 60 and older [50]. Based on these reports and the results of this study, further research may be needed to clarify the relationship between laughter frequency and hypertension. Further studies, including interventions to increase laughter frequency and assess the impact on subjective happiness and blood pressure, seem necessary.

This study found that individuals with poor chewing ability had a higher likelihood of having hypertension. This suggests that the oral environment may play a role in hypertension. A study of older adults aged 70 or older showed that severe gum disease may be positively associated with hypertension in this age group [51]. Additionally, longitudinal studies have linked periodontal disease to the development of metabolic syndrome [52]. These findings suggest that oral health in older adults may be a contributing factor to uncontrolled blood pressure. The results of this study suggest that promoting good oral hygiene could be important in preventing hypertension in older adults.

This study had several limitations. First, being a cross-sectional study, we were unable to establish a causal relationship between hypertension, laughter frequency, and subjective happiness. Second, the study was based on self-administered questionnaires, which could lead to recall bias and difficulty in determining the required frequency of laughter and the level of subjective happiness to prevent hypertension. Third, in this study, hypertension was solely defined by self-reported data. Meanwhile, details on physician’s diagnosis, management, and disease duration remain unclear. According to the literature, self-reported information about hypertension has shown a substantial agreement with medical records [53]. However, other studies have revealed that self-reports may significantly underestimate the prevalence of hypertension [54]. Therefore, it is imperative to acknowledge that hypertension may have been underestimated in this study as well. Finally, the study was conducted on a population of individuals aged ≥65 years; thus, it is uncertain whether the results apply to other age groups.

## 5. Conclusions

This study suggests that having more opportunities to feel happiness may be important in preventing hypertension. However, it is important to exercise caution in overly pursuing an increase in subjective happiness, as focusing too much on subjective happiness can lead to disappointment and negatively impact one’s overall well-being.

## Figures and Tables

**Figure 1 ijerph-20-05713-f001:**
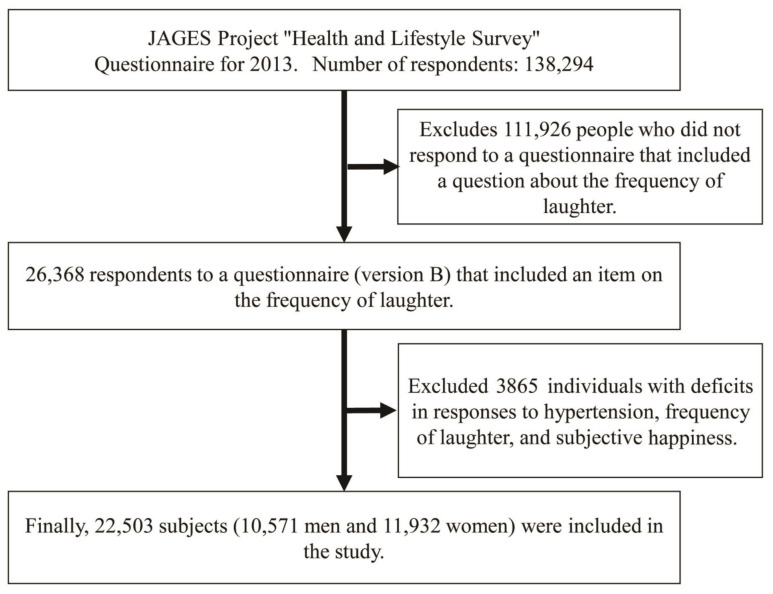
Flowchart of research subject selection.

**Table 1 ijerph-20-05713-t001:** Characteristics of the participants.

		Hypertension			
Factors	Class	Yes	No	Total	*p*-Value
Age (years)		73.3 ± 6.1	74.4 ± 6.2		<0.001
Sex	Male	5771 (47.5)	4800 (46.3)	10,571 (47.0)	0.066
	Female	6368 (52.5)	5564 (53.7)	11,932 (53.0)	
Laughter frequency/subjective happiness	Infrequent laughter/low level of subjective happiness	4044 (33.3)	3608 (34.8)	7652 (34.0)	0.043
	Frequent laughter/low level of subjective happiness	1792 (14.8)	1570 (15.1)	3362 (14.9)	
	Infrequent laughter/high level of subjective happiness	2941 (24.2)	2428 (23.4)	5369 (23.9)	
	Frequent laughter/high level of subjective happiness	3362 (27.7)	2758 (26.6)	6120 (27.2)	
BMI	Underweight	1068 (9.2)	423 (4.3)	1491 (7.0)	<0.001
	Normal	8612 (74.3)	6526 (66.3)	15,138 (70.6)	
	Obesity	1911 (16.5)	2897 (29.4)	4808 (22.4)	
Caregiving/assistance status	Not needed	11,441 (96.9)	9714 (96.4)	21,155 (96.7)	0.038
	Needed but not received	307 (2.6)	316 (3.1)	623 (2.8)	
	Received	61 (0.5)	43 (0.4)	104 (0.5)	
Household income	JPY ≥2 million	7842 (73.5)	6519 (72.6)	14,361 (73.1)	0.176
	JPY <2 million	2827 (26.5)	2455 (27.4)	5282 (26.9)	
Number of teeth	≥20 teeth	6228 (52.6)	5077 (50.3)	11,305 (51.6)	<0.001
	<20 teeth	5605 (47.4)	5011 (49.7)	10,616 (48.4)	
Chewing ability	Anything	4818 (39.8)	3974 (38.5)	8792 (39.2)	0.027
	Almost	6522 (53.9)	5717 (55.3)	12,239 (54.5)	
	Limited	728 (6.0)	595 (5.8)	1323 (5.9)	
	Rarely chewing	28 (0.2)	42 (0.4)	70 (0.3)	
	Liquid diet	10 (0.1)	7 (0.1)	17 (0.1)	
Drinking habits	No drinker	7367 (61.4)	5953 (58.1)	13,320 (59.9)	<0.001
	Former drinker	604 (5.0)	518 (5.1)	1122 (5.0)	
	Current drinker	4019 (33.5)	3771 (36.8)	7790 (35)	
Smoking habits	No smoker	8722 (72.9)	7620 (74.6)	16,342 (73.7)	<0.001
	Former smoker	1875 (15.7)	1700 (16.6)	3575 (16.1)	
	Current smoker	1366 (11.4)	896 (8.8)	2262 (10.2)	
TMIG-IC	Good	9672 (79.7)	8239 (79.5)	17,911 (79.6)	0.737
	Poor	2467 (20.3)	2125 (20.5)	4592 (20.4)	
Frequency of seeing friends	4+ times/week	2048 (17.6)	1710 (17.3)	3758 (17.5)	0.518
	2–3 times/week	2282 (19.6)	2019 (20.4)	4301 (20.0)	
	Once/week	1550 (13.3)	1317 (13.3)	2867 (13.3)	
	1–3 times/month	2640 (22.7)	2159 (21.9)	4799 (22.3)	
	A few times/year	2149 (18.5)	1821 (18.4)	3970 (18.4)	
	Not seeing friends	974 (8.4)	854 (8.6)	1828 (8.5)	
Number of friends met	≥10	4158 (35.7)	3362 (34.1)	7520 (35.0)	0.021
	6–9	1633 (14.0)	1314 (13.3)	2947 (13.7)	
	3–5	2833 (24.3)	2534 (25.7)	5367 (24.9)	
	1 or 2	2067 (17.7)	1778 (18.0)	3845 (17.9)	
	0 (none)	967 (5.5)	866 (5.8)	1833 (5.7)	
Family living arrangement	Others	9974 (85.7)	8475 (85.6)	18,449 (85.7)	0.894
	Living alone	1666 (14.3)	1423 (14.4)	3089 (14.3)	
Employment	Employed	2846 (24.9)	2213 (22.9)	5059 (24.0)	<0.001
	Unemployed	8598 (75.1)	7455 (77.1)	16,053 (76.0)	
Hobbies	Yes	10,395 (87.8)	8664 (86.1)	19,059 (87.0)	<0.001
	No	1444 (12.2)	1401 (13.9)	2845 (13.0)	
Marital status	With spouse (including common-law marriage)	8791 (73.6)	7197 (70.7)	15,988 (72.3)	<0.001
	Others	3148 (26.4)	2981 (29.3)	6129 (27.7)	

Data were expressed as means ± SDs or n (%). Differences between the two groups were analyzed using the chi-square test. BMI, body mass index; TMIG-IC, Tokyo Metropolitan Institute of Gerontology Index of Competence.

**Table 2 ijerph-20-05713-t002:** Results of the age- and sex-adjusted multivariate logistic regression analyses of the association between laughter frequency and subjective happiness and hypertension.

		All		Men	Women
Factors	Class	Age- and Sex- Adjusted OR (95% CI)	Multiple-Adjusted OR (95% CI) ^†^ (n = 22,503)	Multiple-Adjusted OR (95% CI) ^‡^ (n = 10,571)	Multiple-Adjusted OR (95% CI) ^‡^ (n = 11,932)
Age	Continuous	1.03 (1.03–1.03)	1.04 (1.03–1.04)	1.02 (1.01–1.03)	1.05 (1.04–1.06)
Sex	Female (ref. male)	1.04 (0.99–1.10)	1.23 (1.15–1.32)		
Laughter frequency/subjective happiness	Frequent laughter/low level of subjective happiness (ref. Infrequent laughter/low level of subjective happiness)	1.00 (0.92–1.09)	0.99 (0.91–1.08)	0.96 (0.85–1.09)	1.03 (0.91–1.15)
	Infrequent laughter/high level of subjective happiness	0.90 (0.84–0.96)	0.91 (0.84–0.98)	0.85 (0.77–0.94)	0.97 (0.87–1.07)
	Frequent laughter/high level of subjective happiness	0.92 (0.86–0.98)	0.91 (0.85–0.98)	0.90 (0.81–1.00)	0.93 (0.84–1.03)
BMI	Underweight (ref. normal)	0.49 (0.44–0.55)	0.49 (0.44–0.56)	0.44 (0.36–0.54)	0.54 (0.46–0.62)
	Obesity	2.06 (1.93–2.20)	2.06 (1.93–2.20)	1.93 (1.76–2.13)	2.19 (1.99–2.41)
Caregiving/assistance status	Needed but not received (ref. not needed)	1.05 (0.89–1.23)			
	Received	0.69 (0.46–1.02)			
Number of teeth	<20 teeth (ref. ≥20 teeth)	1.00 (0.95–1.06)			
Chewing ability	Almost (ref. anything)	1.02 (0.96–1.08)	1.04 (0.98–1.10)	1.01 (0.93–1.10)	1.04 (0.96–1.13)
	Limited	0.90 (0.80–1.01)	0.92 (0.82–1.04)	0.96 (0.81–1.14)	0.88 (0.73–1.05)
	Rarely chewing	1.64 (1.01–2.65)	1.65 (1.00–2.72)	1.55 (0.80–3.00)	1.71 (0.79–3.69)
	Liquid diet	0.77 (0.29–2.02)	0.85 (0.31–2.29)	1.39 (0.39–5.04)	0.37 (0.07–1.97)
Drinking habits	Former drinker (ref. no drinker)	1.16 (1.02–1.32)	1.15 (1.00–1.31)	1.41 (1.21–1.65)	1.07 (0.81–1.41)
	Current drinker	1.34 (1.25–1.42)	1.38 (1.29–1.47)	1.78 (1.63–1.94)	0.96 (0.87–1.07)
Smoking habits	Former smoker (ref. no smoker)	1.10 (1.01–1.19)	1.04 (0.95–1.13)	1.01 (0.92–1.11)	0.81 (0.62–1.05)
	Current smoker	0.82 (0.75–0.91)	0.82 (0.75–0.91)	0.77 (0.69–0.86)	0.95 (0.76–1.17)
Number of friends met	6–9 (ref. ≥10)	0.98 (0.90–1.07)	0.99 (0.91–1.08)	0.94 (0.82–1.08)	1.03 (0.92–1.15)
	3–5	1.08 (1.01–1.16)	1.08 (1.01–1.17)	1.12 (1.00–1.24)	1.05 (0.95–1.16)
	1 or 2	1.03 (0.95–1.12)	1.02 (0.94–1.11)	1.04 (0.92–1.17)	1.01 (0.90–1.14)
	0 (none)	1.07 (0.96–1.18)	1.06 (0.95–1.18)	1.10 (0.95–1.26)	1.03 (0.86–1.23)
Employment	Unemployed (ref. employed)	1.00 (0.94–1.07)			
Hobbies	No (ref. Yes)	1.11 (1.03–1.20)	1.09 (1.00–1.19)	1.07 (0.94–1.22)	1.10 (0.98–1.23)
Marital status	Others (ref. with spouse (including common-law marriage))	1.04 (0.98–1.11)			

^†^ Adjusted for age, sex, laughter frequency/subjective happiness, BMI, chewing ability, drinking habits, smoking habits, number of friends met, hobbies, and marital status. Logistic regression model; a *p* value of <0.05 was considered statistically significant. ^‡^ Adjusted for age, laughter frequency/subjective happiness, BMI, chewing ability, drinking habits, smoking habits, number of friends met, hobbies, and marital status. Logistic regression model; a *p* value of <0.05 was considered statistically significant. CI: confidence interval, OR: odds ratio, Ref: reference, BMI: body mass index.

**Table 3 ijerph-20-05713-t003:** Results of the multivariate logistic regression analysis of the association between the laughter frequency or subjective happiness and hypertension.

Factors	Class	Multiple-Adjusted OR (95% CI) ^†^ (n = 22,503)	Multiple-Adjusted OR (95% CI) ^‡^ (n = 22,503)
Age	Continuous	1.04 (1.03–1.04)	1.04 (1.03–1.04)
Sex	Female (ref. male)	1.23 (1.15–1.31)	1.23 (1.15–1.32)
Laughter frequency	Frequent laughter (ref. infrequent laughter)	0.98 (0.93–1.04)	
Subjective happiness	High level of subjective happiness (ref. low level of subjective happiness)		0.91 (0.86–0.97)
BMI	Underweight (ref. Normal)	2.03 (1.80–2.29)	2.03 (1.80–2.29)
	Obesity	4.19 (3.68–4.76)	4.18 (3.67–4.75)
Chewing ability	Almost (ref. anything)	1.04 (0.99–1.10)	1.04 (0.98–1.10)
	Limited	0.94 (0.83–1.06)	0.92 (0.82–1.05)
	Rarely chewing	1.69 (1.03–2.78)	1.65 (1.00–2.72)
	Liquid diet	0.85 (0.32–2.31)	0.85 (0.31–2.29)
Drinking habits	Former drinker (ref. no drinker)	1.15 (1.01–1.32)	1.15 (1.00–1.31)
	Current drinker	1.38 (1.29–1.47)	1.38 (1.29–1.47)
Smoking habits	Former smoker (ref. no smoker)	1.04 (0.96–1.13)	1.04 (0.95–1.13)
	Current smoker	0.83 (0.75–0.91)	0.82 (0.75–0.91)
Number of friends met	6–9 (ref. ≥10)	1.00 (0.91–1.09)	0.99 (0.91–1.08)
	3–5	1.09 (1.02–1.18)	1.08 (1.01–1.17)
	1 or 2	1.04 (0.95–1.13)	1.02 (0.94–1.11)
	0 (none)	1.07 (0.96–1.20)	1.06 (0.95–1.18)
Hobbies	No (ref. yes)	1.10 (1.01–1.20)	1.09 (1.00–1.19)

^†^ Adjusted for age, sex, laughter frequency, BMI, chewing ability, drinking habits, smoking habits, number of people met, hobbies, and marital status. Logistic regression model; a *p* value of < 0.05 was considered statistically significant. ^‡^ Adjusted for age, sex, subjective happiness, BMI, chewing ability, drinking habits, smoking habits, number of people met, hobbies, and marital status. Logistic regression model; a value of *p* < 0.05 was considered statistically significant. CI: confidence interval, OR: odds ratio, Ref: reference, BMI: body mass index.

## Data Availability

Additional detailed data for this study are available upon reasonable written request to the corresponding author.

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
