# Peer review of "Subjective Happiness, Frequency of Laughter, and Hypertension: A Cross-Sectional Study Based on the Japan Gerontological Evaluation Study (JAGES)"

_ijerph, 2023, doi:10.3390/ijerph20095713_

Round 1

Reviewer 1 Report

Introduction good and evidence presented from the literature. Laughter explored in relation to hypertension. This was a first study of its kind. Study used data from 22,503 respondents. The methodology clearly explained and the characteristics shown in Table 1. Logistic regression  analysis was used.

This study showed the link between positive emotions and a reduced risk of hypertension. 

The limitations were clearly outlined as it was a cross sectional study there was no causal relationship found between hypertension, laughter and subjective happiness. Recall bias was identified. 

Author Response

Answers to reviewer 1:

Thank you very much for your letter regarding our manuscript. We appreciate your comments, which have helped us to improve our manuscript. The changes were made and our point-by-point responses to the reviewer’s comments were described below.

Comment #1.

Introduction good and evidence presented from the literature. Laughter explored in relation to hypertension. This was a first study of its kind. Study used data from 22,503 respondents. The methodology clearly explained and the characteristics shown in Table 1. Logistic regression analysis was used.

This study showed the link between positive emotions and a reduced risk of hypertension.

The limitations were clearly outlined as it was a cross sectional study there was no causal relationship found between hypertension, laughter and subjective happiness. Recall bias was identified.

Response #1.

Thank you for the valuable comments. As pointed out, the paper was revised accordingly. We hope you can review it again.

Reviewer 2 Report

Few grammatical comments - see attached.

Would have liked to see some discussion on the multivariate logistic regression conducted separately for males and females. Sees like the negative association of happiness with hypertension was only among males.

Author Response

Answers to reviewer 2

Thank you very much for your letter regarding our manuscript. We appreciate your comments, which have helped us to improve our manuscript. The changes were made and our point-by-point responses to the reviewer’s comments were described below.

Comment #1.

Few grammatical comments - see attached.

Response #1.

Thank you for the valuable comments. As pointed out by the reviewer, the uploaded file was reviewed, and the text was revised. Further, information on the association between hobbies and happiness was added (page 10, line 9–page 10, line 23).

Comment#2.

Would have liked to see some discussion on the multivariate logistic regression conducted separately for males and females. Sees like the negative association of happiness with hypertension was only among males.

Response #2.

Thank you for the valuable comment. As pointed out by the reviewer, data related to the analysis results according to sex were added (page 9, lines 13–page 10, line 8).

Reviewer 3 Report

Overall, it is a concisely written paper without any superfluities. Because  of no complexity in the subject, variables, and research methods, there is not any special things to comment on.

Just to ask a few questions

1. The definition of the dependent variable is ambiguous. It is unclear whether this was a doctor's diagnosis or an arbitrary judgment. Since both current and past hypertension were included, but it is unclear how long ago hypertension was included. In addition, among past hypertension, whether only unmanaged hypertension is included, or if the current status is controlled and not diagnosed as hypertension, and if there is a medical history of hypertension in the past, is it included in the dependent variable?  

 2. As the authors mention, the limitations of the self-administered survey, rather than the interview survey, seem clear. Since it was a self-administered questionnaire for the elderly, it is questionable whether the intention of the questionnaire was accurately identified and answered.

Author Response

Answers to reviewer 3

Thank you very much for your letter regarding our manuscript. We appreciate your comments, which have helped us to improve our manuscript. The changes were made and our point-by-point responses to the reviewer’s comments were described below.

Comment#1.

The definition of the dependent variable is ambiguous. It is unclear whether this was a doctor's diagnosis or an arbitrary judgment. Since both current and past hypertension were included, but it is unclear how long ago hypertension was included. In addition, among past hypertension, whether only unmanaged hypertension is included, or if the current status is controlled and not diagnosed as hypertension, and if there is a medical history of hypertension in the past, is it included in the dependent variable?

Response #1.

As pointed out by the reviewer, the definition of hypertension in this study was solely based on self-reported information, and data related to medical diagnosis or management or disease duration remain unclear. Consequently, information on underestimation regarding the confirmation of hypertension was added in the Discussion section (page 11, lines 3–10).

Comment#2.

As the authors mention, the limitations of the self-administered survey, rather than the interview survey, seem clear. Since it was a self-administered questionnaire for the elderly, it is questionable whether the intention of the questionnaire was accurately identified and answered.

Response #2.

The reviewers’ concerns are valid. However, several studies have examined the validity of some JAGES questionnaires (Saito T., et al. Geriatr Gerontol Int. 2019. Watanabe R., et al. Geriatr Gerontol Int. 2022). Therefore, the authors believe that the study results should be validated.

Reviewer 4 Report

The problem of hypertension is a global problem and its solution requires the consolidated efforts of physicians, social workers and psychologists. The presented study attempts to study the relationship between Subjective happiness, frequency of laughter, and hypertension. The authors proceed from the idea of the importance of social factors in the development of hypertension. Judging by the latest data from Russian researchers on "positive aging", such a connection is quite predictable. However, the value of this study lies in clarifying a number of behavioral, hygienic and relational (social) factors (other than happiness) in the development of hypertension.

Some wishes. The introduction needs to be expanded in order to theoretically substantiate the studied connection. In the empirical part of the article and in the discussion, I would like a clearer picture of what part of the variance is explained by the analyzed variables in men and women, because the presence of just some kind of connection does not bother anyone. Finally, the specification of the initial hypotheses is required.

I wish the authors success in the subsequent stages of the publication of the article.

Author Response

Answers to reviewer 4

Thank you very much for your letter regarding our manuscript. We appreciate your comments, which have helped us to improve our manuscript. The changes were made and our point-by-point responses to the reviewer’s comments were described below.

Comment#1.

The introduction needs to be expanded in order to theoretically substantiate the studied connection.

Response #1.

Thank you for the valuable comment. As suggested by the reviewer, additional data were added in the Introduction section (page 2, lines 17–29).

Comment#2.

In the empirical part of the article and in the discussion, I would like a clearer picture of what part of the variance is explained by the analyzed variables in men and women, because the presence of just some kind of connection does not bother anyone.

Response #2.

Thank you for the valuable comments. As instructed by the reviewer, data on the analysis results based on sex were added (page 9, line 13–page 10, line 8).

Comment#3.

Finally, the specification of the initial hypotheses is required.

Response #3.

Thank you for the valuable comments. As pointed out by the reviewer, information on the first hypothesis was added to the Introduction section (page 2, lines 35–37).